# Flooding and ecological restoration promote wetland microbial communities and soil functions on former cranberry farmland

**Rachel L. Rubin**[1], **Kate A. Ballantine**[1], **Arden Hegberg**[2], **Jason P. Andras**[2]*

**1** Department of Environmental Studies, Mount Holyoke College, South Hadley, Massachusetts United States of America, **2** Department of Biology, Mount Holyoke College, South Hadley, Massachusetts, United States of America

* jandras@mtholyoke.edu

**Data Availability Statement:** Metadata is attached in the supplementary material. Sequence files are available under the NCBI sequence read archive (accession #PRJNA693513), and under JGI's

## Abstract

Microbial communities are early responders to wetland degradation, and instrumental players in the reversal of this degradation. However, our understanding of soil microbial community structure and function throughout wetland development remains incomplete. We conducted a survey across cranberry farms, young retired farms, old retired farms, flooded former farms, ecologically restored former farms, and natural reference wetlands with no history of cranberry farming. We investigated the relationship between the microbial community and soil characteristics that restoration intends to maximize, such as soil organic matter, cation exchange capacity and denitrification potential. Among the five treatments considered, flooded and restored sites had the highest prokaryote and microeukaryote community similarity to natural wetlands. In contrast, young retired sites had similar communities to farms, and old retired sites failed to develop wetland microbial communities or functions. Canonical analysis of principal coordinates revealed that soil variables, in particular potassium base saturation, sodium, and denitrification potential, explained 45% of the variation in prokaryote communities and 44% of the variation in microeukaryote communities, segregating soil samples into two clouds in ordination space: farm, old retired and young retired sites on one side and restored, flooded, and natural sites on the other. Heat trees revealed possible prokaryotic (*Gemmatimonadetes*) and microeukaryotic (*Rhizaria*) indicators of wetland development, along with a drop in the dominance of *Nucletmycea* in restored sites, a class that includes suspected mycorrhizal symbionts of the cranberry crop. Flooded sites showed the strongest evidence of wetland development, with triple the soil organic matter accumulation, double the cation exchange capacity, and seventy times the denitrification potential compared to farms. However, given that flooding does not promote any of the watershed or habitat benefits as ecological restoration, we suggest that flooding can be used to stimulate beneficial microbial communities and soil functions during the restoration waiting period, or when restoration is not an option.

Genome Portal (projects 1191228 & 1191229) (https://genome.jgi.doe.gov/portal/Theeffcowetlands/Theeffcowetlands.info.html).

**Funding:** Funding was provided by Joint Genome Institute Project 1188787. The funder had no role in study design, data collection, decision to publish, or preparation of the manuscript.

**Competing interests:** The authors have declared that no competing interests exist.

## Introduction

A paradigm shift has occurred in society's perspective on wetlands. The "drain the swamp" mentality of the early 1900's contrasts with today's rallying calls to preserve, restore, and rewild. In recognition of the role of wetlands for critical ecosystem services, such as carbon storage and nitrogen removal, the United Nations declared 2021–2030 the decade on ecological restoration. While progress is evident, science is still unclear on how to best restore wetlands because the return of plant and animal communities does not guarantee the return of desirable ecosystem functions [1, 2]. Soil microbial community surveys can be used to gauge whether treated sites are on track to meet restoration goals, particularly those that accrue slowly and are difficult to measure, such as soil organic matter accumulation and nitrogen removal.

The physical soil environment influences microbial community composition, which in turn influences the physical soil environment. Despite this established feedback loop, it is still unclear whether discrete community types align closely with soil biogeochemistry (indicator species; [3–5]), or whether community members are redundant to one another, their identity uncoupled from their biogeochemistry [6, 7]. Evidence to support these theories remains incomplete, with much of this research effort focusing on prokaryotes (bacteria and archaea). Microeukaryotes—the microscopic fungi, soil animals, and protists that occur in high numbers in soil and water, have received less attention. Characterizing prokaryotes and microeukaryotes together should yield a more complete picture of the relationships between land management, soil physicochemical gradients, and microbial community structure. Within this field of inquiry, restored peat wetlands are a particularly underexplored ecosystem, and evaluating how microbial communities align with soil biogeochemistry should also provide insights for managing these unique ecosystems.

Cranberries (*Vaccinium macrocarpon*) are native to New England and occur naturally in kettle hole bogs and marshes. Indigenous groups including the Wampanoag have been cultivating and harvesting cranberries for over 12,000 years before commercial cranberry farming spread rapidly across the region in the 1800's. At least half of Massachusetts cranberry farms are built on existing peat wetlands [8], exploiting natural springs for irrigation and harvesting purposes. On modern cranberry farms, water is supplied through a system of dams, ditches, and culverts. Pesticides are applied regularly, and thin layers of sand, often accumulating up to a meter in depth, are applied to promote drainage and prevent root dieback [9]. Until the 1990's, Massachusetts was the worldwide leader in total harvestable acres, but due to climate and economic factors, at least 40% of existing farmland will be retired in the next decade [8].

The cranberry retirement wave presents a widespread opportunity, both for wetland restoration and for research on ecosystem trajectories. Active ecological restoration on former cranberry farms facilitates ecosystem recovery through earth moving, ditch filling, surface grading, and channel or floodplain reconstruction. When no action is taken, retired farms are typically drained of existing water. Over time, it appears that retired sites will often transition to an upland maple or pine forest, never regaining their historical wetland status [10].

Prior to restoration, cranberry farms are typically placed into conservation easements, and it is not uncommon for farms to be retired for 15 years or longer before they can be restored. Rather than allowing retired farms to develop into upland forests, the USDA National Resources Conservation Service (NRCS) experimented with leaving retired farms permanently flooded, in the harvest condition. While flooding clearly maintains ponded water and wet soils, the soil characteristics and microbial communities of flooded sites have not yet been characterized. If flooded sites have comparable soil-based benefits as restored sites, then flooding could be used as a mitigation strategy for sites awaiting restoration, or as an alternative when restoration is not an option.

We conducted a survey of soil microbial communities and ecosystem functions related to soil development and nutrient cycling across 23 sites and five treatments (farm, young retired, old retired, flooded, and restored) in southeastern Massachusetts. Our research questions were: 1) Which of the five treatments had the most similar microbial communities to natural sites? 2) How much variation in community composition can be explained by soil variables, and which of these variables align most closely to microbial community composition? 3) Which taxa are indicators of wetland development? Given the results of our previous work [11], we hypothesized that restored sites would harbor similar microbial communities to natural sites. While we had not examined flooded sites before, we hypothesized that flooded sites would also have similar characteristics to natural wetlands, but to a lesser extent than restored sites, due to their permanently flooded (unnatural) hydrologic regime. We expected that denitrification potential and soil organic matter would be important for explaining variation in microbial communities across the six treatment categories, and that these variables would be highest at the wet sites (natural, restored, and flooded) and lowest at the drier sites (farm, young retired, and old retired).

## Methods

### Field sampling and site histories

We sampled twenty-three field sites across southeastern Massachusetts during July 2017, spanning a perimeter of 100 km and total area of 500 km2. Treatments included: cranberry farm (3 sites), young retired (5 sites), old retired (3 sites), flooded (3 sites), restored (6 sites), and natural sites (3 sites) (Fig 1). Natural sites have no history of cranberry farming. Restored sites, flooded sites and young retired sites were roughly comparable in age; restored sites had been restored either 1 year (Tidmarsh) or 6 years (Eel River) prior to sampling, flooded sites were flooded 4 years prior to sampling, and young retired sites were retired 1–4 years prior to sampling. Old retired sites were retired 17–20 years prior to sampling, and farms had been continuously farmed for at least 50 years. In our study, "treatments" include multiple sites with varying site ages, and conclusions are drawn with this in mind.

The flooded sites known as "Grassi", "Goldawitz" and "Pembroke" are NRCS conservation easement sites. Early in the Wetlands Reserve Program, the agency made use of existing water control structures (flumes and wooden boards) to flood the farms and create artificial wetlands (S1 Fig in S1 File). Ecologically restored sites were manipulated according to a methodology developed by the Massachusetts Department of Environmental Restoration (DER). Procedures include removal of dams and water control structures, plugging irrigation ditches with sand, surface roughening and microtopography grading, and, when needed, channel and floodplain reconstruction. No seeding was performed at either the restored or the flooded sites, and previous work has shown that native plants recolonize these sites through activation of the belowground seedbank [12, 13].

Some sampling sites were clustered together whereas others were more spread out (Fig 1). This sampling scheme was deployed to capture variation within particularly large restoration projects (Tidmarsh West and Eel River), or when sites co-occurred with other outside long-term monitoring objectives (Tidmarsh East). To validate that clustered sampling still captured more variation across these areas, we conducted a sensitivity test by systematically dropping all possible combinations of two sites from Tidmarsh West, Eel River and Tidmarsh East, thereby equalizing site replication across treatment categories (S2 Fig in S1 File). For the majority of cases (8 cases out of 12), dropping two sites at a time from Tidmarsh East (1, 2, and 3), Tidmarsh West (1, 2, and 3) and Eel River (North, Northeast, and South) decreased variation rather than increased variation. Therefore, we retained all 23 sites throughout subsequent analyses.

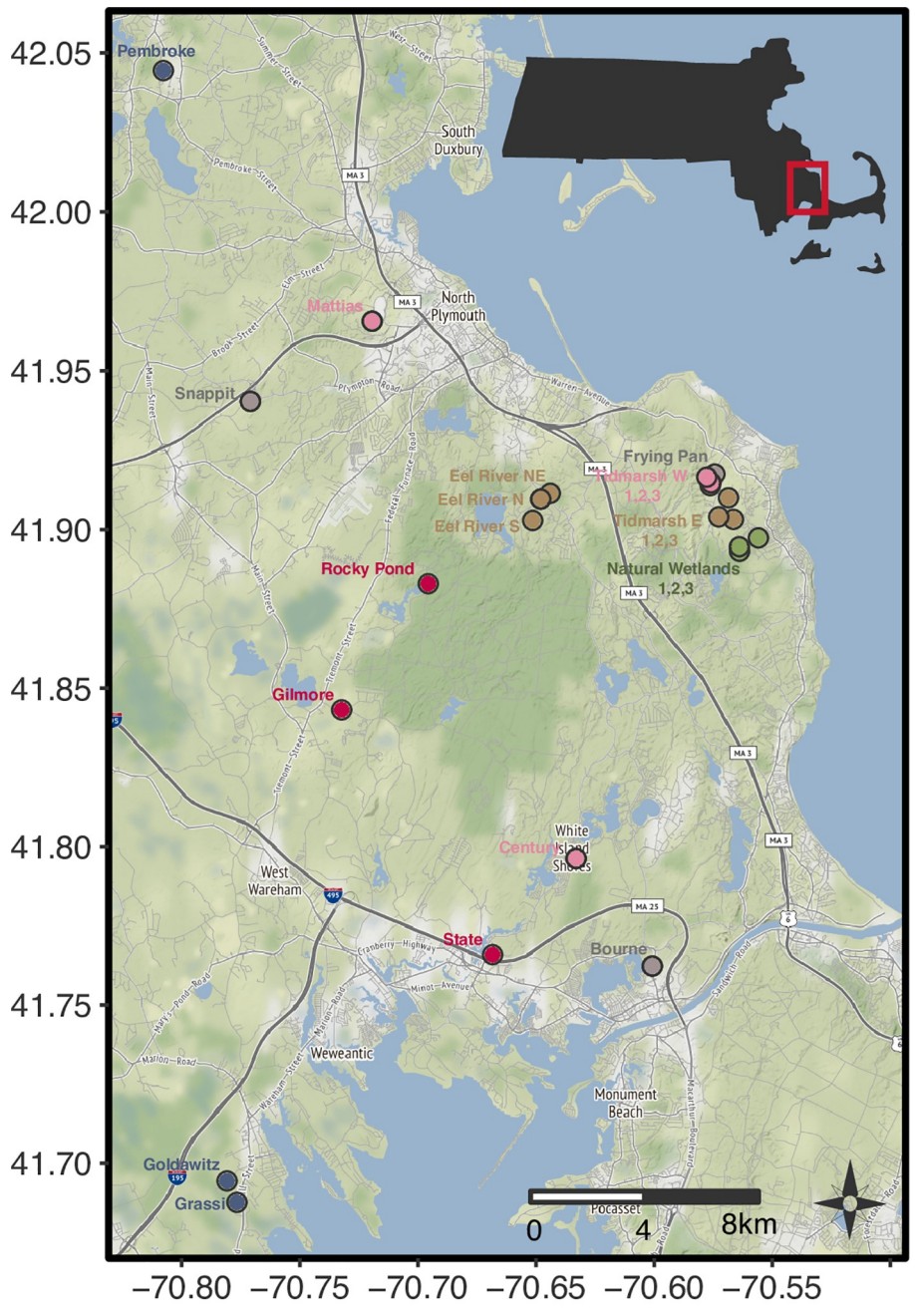

**Fig 1. Map of study area.** 23 sites were surveyed in southeast Massachusetts, spanning five treatments. Map tiles by Stamen Design, under CC BY 3.0. Data by OpenStreetMap (WGS-84), under ODbL.

Eight soil samples (dysic, mesic Typic Haplosaprists of the Freetown series), were randomly selected at each site, spaced 30 m apart. Plants and mosses were removed from the soil surface, and soils were collected using a 1.5-cm diameter soil corer to a depth of 10 cm, and flame sterilized between each sample. Soil samples were homogenized in their own bags and subsampled in the field: half of each sample was flash frozen in liquid nitrogen and stored at -80°C until DNA extraction, and the other half was stored at 5°C for the remaining 35 soil analyses.

## DNA sequencing and bioinformatics

DNA was extracted using a DNeasy Powersoil Kit (QIAGEN Inc, Hilden DE) using 0.2 g of soil. For prokaryotic amplicons, libraries were prepared by amplifying the V4-V5 hypervariable region of the 16S rRNA gene using primers 515F (5- GTGYCAGCMGCCGCGGTAA-3), and 926R (5-CCGYCAATTYMTTTRAGTTT-3) [14] and sequenced at the Joint Genome Institute (Walnut Creek, CA, USA) on an Illumina MiSeq 96 cycle cartridge, across four runs. For eukaryotic amplicons, libraries were prepared by amplifying the V9 hypervariable region of the 18S rRNA gene using primers 1391F (5-GTACACACCGCCCGTC-3) and reverse primer EukBR (5-TGATCCTTCTGCAGGTTCACCTAC-3) [15, 16] and sequenced at Argonne National Lab on an Illumina MiSeq 300 cycle V2 cartridge, on a single run.

Raw sequences were processed using Quantitative Insights into Microbial Ecology 2 (QIIME2) [17]. Amplicon sequence variants (ASVs) were identified using the Divisive Amplicon Denoising Algorithm (DADA2), which merges sequences, removes chimeric sequences, removes reads with ambiguous bases, and corrects sequence errors so that amplicon sequence variants can be resolved to the level of single-nucleotide differences [18].

For 16S amplicons, reverse reads were low quality (this is common for this gene region), so only forward reads were used. Sequences were truncated to 277 base pairs, and denoising was performed separately on each run, resulting in 22,176,127 total sequences (an average of 129,857 sequences per sample), and 13,978 unique ASVs across 183 samples (an average of 1,579 ASV's per sample). One sample, TE3.2, was removed at the denoising step due to low sequence count. Feature tables and representative sequence files from each of the four runs were merged, and a denovo phylogenetic tree was made using the align-to-tree-mafft-fastree pipeline. Reads from the V4-V5 hypervariable region were extracted from the SILVA ver. 132 database [19], and a feature classifier was trained on that gene region. Taxonomy was assigned at 99% similarity using the SILVA ver. 132 QIIME2 release. Following taxonomy assignment, sequences and feature tables were filtered to remove mitochondrial and chloroplast sequences, as well as sequences that were unassigned at the domain level.

For 18S amplicons, forward and reverse reads were trimmed to 144 bp and denoised, resulting in 11,430,013 total sequences (average of 36,859 sequences per sample) amounting to 70,836 unique ASVs across 183 samples (average of 407 ASVs per sample). One sample (noted as M.3 in the **S1 Data**) was removed during denoising due to low sequence count. Phylogenetic trees and feature classification were performed in a similar manner as the 16S analyses, this time using a feature classifier trained on the V9 hypervariable region of the 18S gene. Taxonomy was assigned at 99% similarity, and sequences and feature tables were filtered to remove plants and unassigned sequences at the domain level.

## Soil variables

We measured 35 soil variables to characterize the soil physicochemical environment (**S1 File**). Soil analyses were performed by the University of Massachusetts Plant and Nutrient Testing Laboratory, using standard protocols. Macro and micronutrient concentrations were determined using the Modified Morgan extraction procedure [20]. Cation exchange capacity was determined using hydrochloric acid cation displacement method [21], and base saturation of calcium, magnesium, and potassium was determined using atomic absorption spectrometry. Soil organic matter was determined through loss on ignition at 360˚ C.

Denitrification potential was measured at the Cary Institute of Ecosystem Studies using the denitrification enzyme assay method [22]. This method adds an abundance of carbon and nitrate to each sample in an oxygen free container, and uses an acetylene inhibitor to prevent reduction of $N_2O$ to $N_2$. The amount of $N_2O$ produced is proportional to the amount of

denitrification enzyme present. Potential methane emissions were measured following ten-day soil incubations, and were quantified using a Shimadzu GC-8 gas chromatograph with a flame ionization detector. Detailed methods for all soil variables are available in **S1 File**.

## Statistical analyses

QIIME2 feature tables, representative sequences, taxonomy tables, and phylogenetic trees were imported into R (v. 1.2.1335, R Core Team 2020) using the qiime2R package [23]. Following conversion to phyloseq objects, we removed samples with less than 1000 sequences, which dropped 14 additional samples from the prokaryote dataset and no additional samples from the microeukaryote dataset.

Spatial dependency is a familiar challenge in microbial studies, and occurs in two ways: 1) environmental gradients are spatially structured, causing spatial structuring of taxa, or 2) biological processes such as speciation, extinction, dispersal or species interactions are distance-related. To test for the first type of spatial dependency, we compared the Euclidean distance matrix of GPS coordinates against the Euclidean distance of soil variables (Wisconsin double-transformed to control for different measurement scales). To test for the second type of spatial dependency, we compared the Euclidean distance matrix of GPS coordinates against the weighted UniFrac microbial community distance matrix. We performed both analyses across the whole dataset, and within each treatment category. There was no relationship between geography and soil variables in either case. In comparing geography and community composition, there was no autocorrelation for prokaryotes, but there was significant autocorrelation for microeukaryotes. We controlled for spatial autocorrelation when possible in further analyses, but acknowledge that spatial relationships cannot be fully decoupled from the treatments, which were also spatially structured (**Fig 1**).

To assess microbial community similarity to natural sites, we used a phylogenetically meaningful distance measure, weighted UniFrac, because microbial phylogeny is highly correlated with functional traits [24]. Weighted UniFrac incorporates the proportion of shared branch lengths amongst the total branch length, as well as the relative abundance of each taxon [25]. Prior to distance calculations, we resolved multichotomies to dichotomies [26] and used a proportional transformation to turn ASV counts into relative abundance, in lieu of rarefaction [27]. We compared unweighted and weighted UniFrac measures for the 16S and 18S datasets, and chose weighted UniFrac for the final presentation because it produced the best model fit ($R^2$) for prokaryotes and microeukaryotes in the canonical analyses of principal coordinates (CAP; methods to follow).

We conducted a similar analysis for the 35 soil variables. For this analysis, we used a Canberra (weighted Manhattan) distance matrix, calculated from Wisconsin double transformed data to account for different measurement scales for each soil variable. We chose Canberra distance rather than the more commonly used Euclidean distance because our dataset had several zeros (values that were below instrument detection limits) and Canberra distance is less sensitive to shared zeros than Euclidean distance. In this particular case, all distance values happened to fall between 0 and 1, so we converted these numbers to similarity (1-dissimilarity) to ease interpretation. We caution that this can only be done when similarity scores fall between 0 and 1, and similarity scores from this study and should not be compared to similarity scores from other studies. For each of the three analyses described above (**Fig 2A–2C**), we also conducted linear mixed effects models on similarity scores using the lme4 package [28], with treatment coded as fixed effect and site included as random effect to control for non-independence of technical replicates taken from the same site (a square-root transformation was applied when necessary to achieve homogeneity of residuals). When omnibus tests were significant ($\alpha$

= 0.05), pairwise comparisons between treatments were estimated using the glht function in the multcomp package [29].

To visualize community differences and to identify the soil variables that relate most strongly to microbial community composition, we used a constrained ordination approach using all 183 technical replicates (canonical analysis of principal coordinates; capscale function in phyloseq) [30, 31]. Like canonical correlation analysis (CCA), CAP can be used with any distance measure and can also remove variance from nuisance variables (we removed latitude and longitude). Unlike CCA, which assumes a unimodal distribution of species along gradients, CAP assumes a linear distribution of species along gradients, which is well suited to environmental gradients at the regional scale (i.e. southeastern Massachusetts). Using weighted UniFrac distance measures, we used a model selection approach to maximize model fit ($R^2$). Redundant covariates were removed, as were covariates that contributed a variance inflation factor of greater than 10. Forward selection was run on soil variables using the ordiR2step function with 200 permutations [32]. Through this process, we achieved a final model with the highest explanatory power and the least number of variables for each of the 16S and 18S datasets. To visualize how environmental gradients align with communities, the top six continuous predictors were plotted as directional vectors.

To visualize differences in within-sample dominance of prokaryotic and microeukaryotic taxa, we constructed differential heat trees using the metacoder package [33]. This method expands on the heat map concept by incorporating both the differential abundance and the hierarchical organization of each taxon. We used the log2 median ratio to compare the differential abundance of each taxon, using natural wetlands as the universal reference group to compare against the other treatments. For instance, the median abundance for a given bacterial class across all the natural wetland samples was divided by the median abundance of that same bacterial class across all the farm samples, and a log2 transformation was applied to normalize the data. Finally, a Wilcoxon Rank-Sum test was used, such that significant differences were shown in color, and non-significant comparisons were shown in grey.

## Results

### Question 1: Which treatments had the most similar microbial communities to natural wetlands?

For prokaryote communities, flooded sites and restored sites had the highest median phylogenetic similarity to natural sites, followed by old retired sites, young retired sites, and farms, which had the lowest similarity (**Fig 2A**). The same pattern held for microeukaryote communities; restored sites and flooded sites had the highest median phylogenetic similarity to natural wetlands, followed by young retired sites, old retired sites, and farm sites (**Fig 2B**). In a similar fashion, the multivariate analysis of all 35 soil variables revealed that flooded sites were the most similar to natural sites, followed by restored sites (**Fig 2C**). Young retired sites, old retired sites and farms had equally lower similarity to natural sites (**Fig 2C**).

### Question 2: How much variation in community composition can be explained by soil variables, and which of these variables have the strongest association with microbial community structure?

Canonical analysis of principal coordinates revealed microbial community differences and the soil variables that explained the greatest variation. The six strongest soil predictors of prokaryote community composition were: potassium base saturation, sodium, zinc, cation exchange capacity, denitrification potential, and iron. Potassium base saturation and sodium structured

## Prokaryotes (16S amplicons)

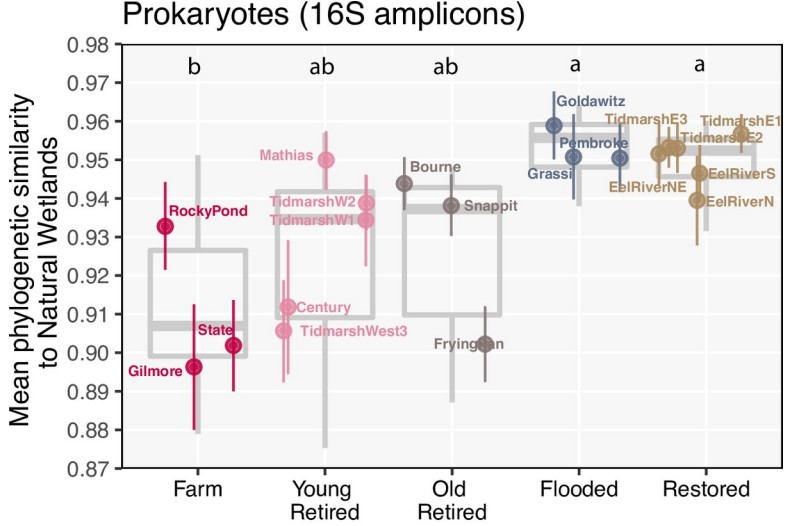

## Microeukaryotes (18S amplicons)

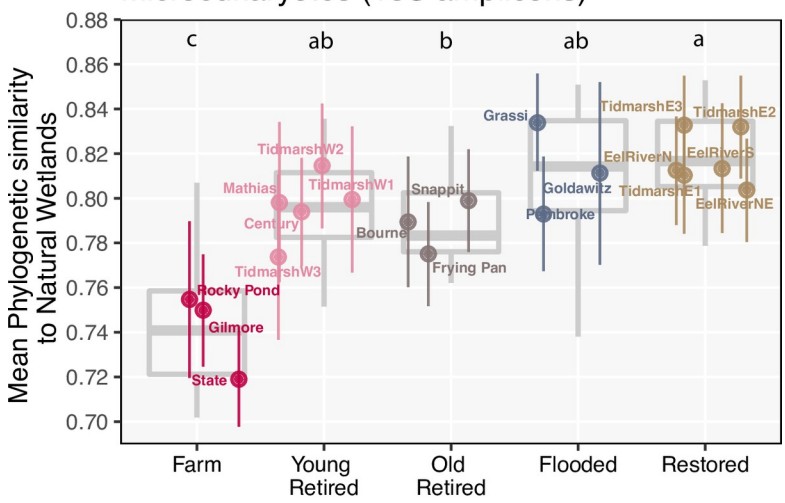

## Soil Variables

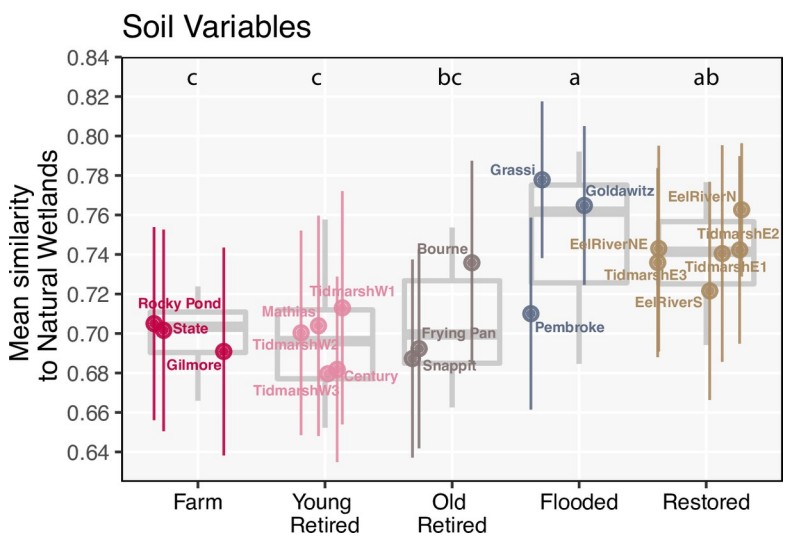

**Fig 2. Similarity between treatments and natural wetland sites.** Plots depict similarity scores for (A) prokaryote communities; (B) microeukaryote communities; and (C) Soil variables. Points indicate the mean similarity of each site to all samples drawn from natural sites, and vertical error bars are standard deviations of similarity scores (eight technical replicates are represented). Grey box plots indicate the median and interquartile range of all soil samples for each treatment category.

communities along CAP axis 1 (**Fig 3A**), in which farm sites, young retired sites and old retired sites had a higher percent potassium base saturation, and restored and flooded sites had higher sodium. Cation exchange capacity was also a large predictor, structuring communities along CAP axis 1 and CAP axis 2. Together, soil variables explained 45% of the variation of the weighted UniFrac distance, whereas 52% of the variation was unconstrained. Conditional variables (latitude and longitude) explained the remaining 3% of the variation in this dataset.

For microeukaryotes, the six largest predictors of community composition were: potassium base saturation, sodium, pH, denitrification potential, copper and phosphorus. Similar to the prokaryotes, microeukaryote communities in farmed, young retired, and old retired sites were correlated with potassium base saturation and sodium along CAP axis 1 (**Fig 3B**), whereas the higher pH at restored and passively restored sites was correlated with communities at restored sites. Together, soil variables explained 44% of the variation in the weighted UniFrac distance, whereas 49% of the variation was unconstrained. Conditional variables (latitude and longitude) explained the remaining 7% of the variation in this dataset.

The twelve soil variables that were identified as important in the CAP analysis also varied significantly across treatments (**Table 1**). Three variables important to wetland development—denitrification potential, cation exchange capacity, and soil organic matter—were highest in flooded sites compared to farm sites, young retired sites, and restored sites. However, we note that that the difference in denitrification potential between flooded sites and restored sites was not statistically significant, and all treatments had less than 12% of the soil organic matter found in natural sites.

## Question 3: Which taxa are indicators of wetland development?

We used heat trees to visualize how the within-sample dominance of taxa differed between treatments and natural wetlands. For prokaryotes, *Parcubacteria* and *Methanobacteria* were more dominant in natural wetlands, shown in teal (**Fig 4**), whereas *Actinobacteria* and *Bacteroidia* were typically less dominant in natural wetlands, shown in orange. We identified a possible indicator of wetland development in the phylum *Gemmatimonadetes*, which was more dominant in natural sites when compared to the farm and retired sites (shown in teal), but equally dominant when compared to flooded sites and restored sites (shown in grey). Another possible indicator was a decline in FCPU426, which was more dominant in farm sites and retired sites when compared to natural sites (shown in orange), but equally dominant when compared to flooded sites and restored sites (shown in grey). Generally speaking, heat trees for farm sites and retired sites were similar (top row), and heat trees for flooded sites and restored sites were similar (bottom row), corroborating our earlier findings from the CAP analysis.

For microeukaryotes, natural sites consistently lacked *Nucletmycea*, a superclass that includes the classes *Fungi* and *Cristidicoidea* (**Fig 5**). While *Nucletmycea* was a clear indicator of cranberry farming and its legacy, *Rigifilida* and *Metamonada* were consistently more dominant in natural wetlands compared to all other treatments. *Rhizaria* was a possible indicator of wetland development; it was more abundant in natural sites, but differences in abundance were much smaller when compared to flooded sites and restored sites (indicated by muted teal and orange shading). Generally speaking, heat trees for farm sites and retired sites were similar

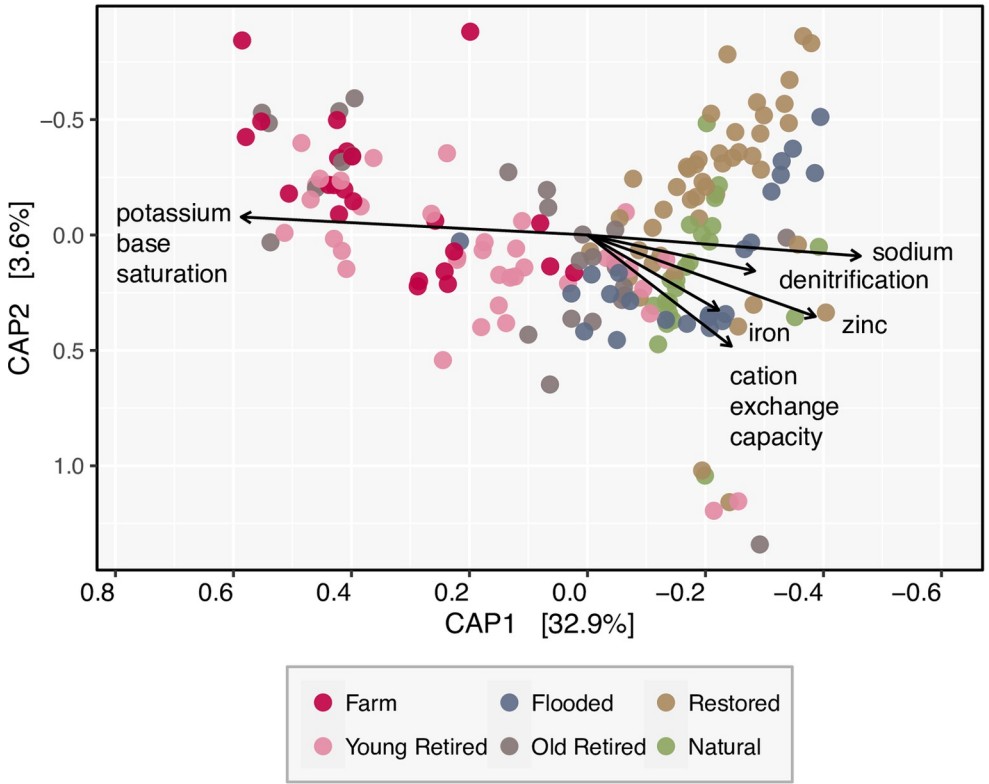

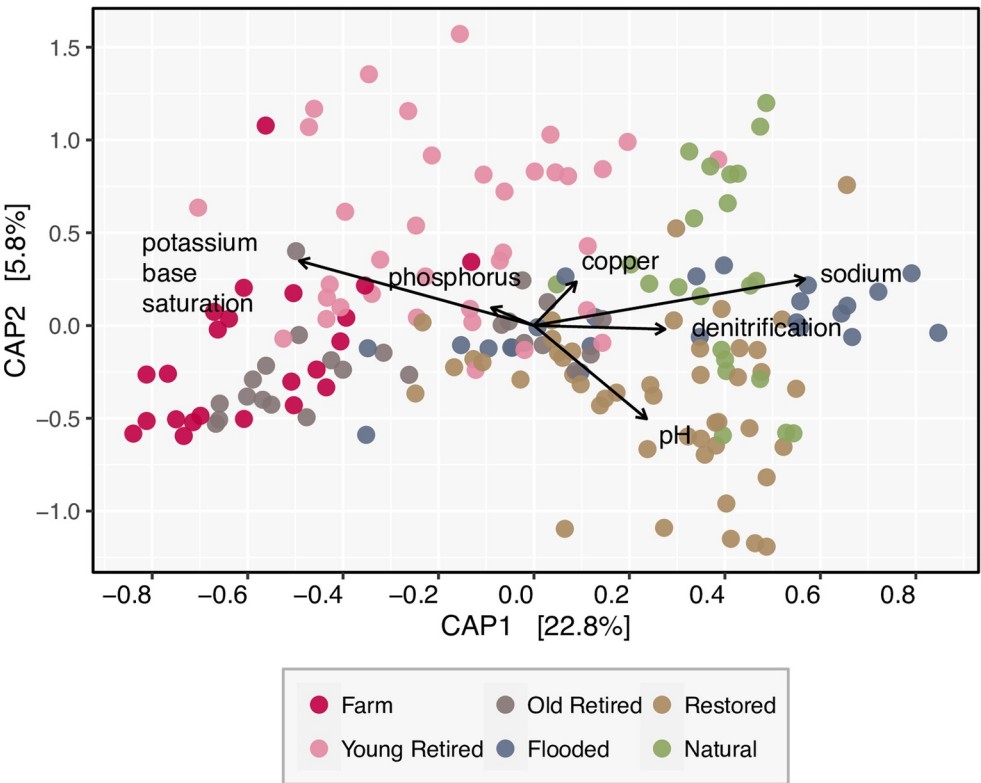

**Fig 3. Canonical analysis of principal coordinates.** (A) Prokaryotic communities were distinct (capscale, p < 0.01), and the six largest predictors of community composition were: potassium base saturation, sodium, zinc, cation exchange capacity, denitrification potential, and iron. Together, soil variables explained 45% of the variation of the weighted UniFrac distance, whereas 52% of the variation was unconstrained. Conditional variables (latitude and longitude) explained the remaining 3% of the variation in this dataset. (B) Microeukaryote communities were distinct (capscale, p<0.01), and the six largest predictors of community composition were: potassium base saturation, sodium, pH, denitrification potential, copper and phosphorus. Together, soil variables explained 44% of the variation in the weighted UniFrac distance, whereas 49% of the variation was unconstrained. Conditional variables (latitude and longitude) explained the remaining 7% of the variation in this dataset.

(top row), and heat trees for flooded sites and restored sites were similar (bottom row), corroborating our earlier findings from the CAP analysis.

We further examined which taxa contributed to the class level patterns detected above. While no prokaryotic genera showed any particular trends during data exploration, a strong pattern appeared within the eukaryote dataset: the fungal genera *Archaeorhizomyces* and *Cairneyella* were the two most abundant eukaryotic genera across the dataset; together, these genera occupied an average of 36% of the total microeukaryote gene abundance per sample, and up to 78% of the total microeukaryote community in one sample. These genera were dominant in cranberry farms, and decreased markedly in retired, restored and natural wetlands (Fig 6).

## Discussion

Restoration success is often defined by the convergence of biological and functional variables towards natural reference conditions. Ecological restoration aims to achieve this by filling in drainage ditches of retired cranberry farms and breaking up the sand layer through surface roughening, thereby retaining water, slowing decomposition, and facilitating soil organic matter accumulation on top of the sandy agricultural substrate. We compared ecological restoration to the retired condition and the permanently flooded condition, which is achieved by retaining the agricultural dams that were formerly used for the cranberry harvest. We found that flooding was remarkably effective at jumpstarting wetland biological and functional

**Table 1. Top twelve soil variables that were identified as important predictors of communities of either prokaryotes or microeukaryotes, as determined by canonical analysis of principal coordinates, along with soil moisture.**

| Variable | Farm | New Retired | Old Retired | Flooded | Restored | Natural | p-value[2] |
|---|---|---|---|---|---|---|---|
| Cation exchange capacity (meq 100 g$^{-1}$ soil) | 6.44 (1.75)[b] | 10.57 (3.09)[b] | 8.75 (2.53)[b] | 12.48 (1.65)[b] | 9.02 (3.46)[b] | 24.84 (12.60)[a] | <**0.001** |
| Copper content (ppm) | 0.11 (0.02) | 0.77 (1.16) | 0.70 (0.95) | 0.78 (0.44) | 0.19 (0.15) | 1.00 (0.63) | ns |
| Denitrification potential (ng N g dry soil$^{-1}$ hour$^{-1}$) | 6.2 (12.3) | 51.1 (131.3) | 124.9 (253.6) | 456.5 (727.3) | 256.4 (423.8) | 883.2 (1,519.8) | ns |
| Iron content (ppm) | 42.75 (22.75) | 67.45 (41.87) | 68.43 (56.34) | 138.01 (49.20) | 107.5 (68.32) | 561.55 (855.79) | ns |
| Magnesium base saturation (%) | 3.61 (0.65) | 3.36 (1.29) | 2.32 (1.02) | 3.63 (1.14) | 4.34 (1.76) | 9.61 (8.18) | ns |
| Methane (ug CH$_4$ g dry soil$^{-1}$ hour$^{-1}$) | 0.01 (0.00)[b] | 0.02 (0.01)[b] | 0.01 (0.00)[b] | 0.06 (0.14)[ab] | 0.03 (0.04)[b] | 0.29 (0.84)[a] | <**0.001** |
| pH | 4.57 (0.19)[b] | 4.32 (0.25)[b] | 4.53 (0.38)[b] | 4.67 (0.52)[ab] | 4.95 (0.27)[a] | 4.17 (0.48)[bc] | <**0.001** |
| Phosphorus content (ppm) | 3.84 (1.62) | 5.59 (2.03) | 3.68 (1.55) | 4.56 (2.10) | 3.43 (2.72) | 4.03 (2.00) | ns |
| Potassium base saturation (%) | 1.94 (0.39)[a] | 1.47 (0.59)[a] | 0.86 (0.31)[b] | 0.85 (0.33)[b] | 0.76 (0.32)[b] | 1.16 (0.46)[ab] | <**0.001** |
| Sodium content (ppm) | 10.80 (3.78)[b] | 11.58 (5.19)[b] | 8.35 (5.89)[b] | 51.33 (62.77)[b] | 19.69 (11.27)[b] | 90.47 (41.16)[a] | <**0.001** |
| Soil moisture (%) | 19.1 (6.6)[b] | 24.4 (9.7)[b] | 14.0 (10.2)[b] | 34.7 (16.9)[b] | 36.6 (12.7)[b] | 74.4 (21.4)[a] | <**0.001** |
| Soil organic matter (%) | 2.22 (0.89)[b] | 3.87 (2.20)[b] | 2.71 (1.42)[b] | 6.02 (2.80)[b] | 3.96 (2.81)[b] | 50.76 (33.45)[a] | <**0.001** |
| Zinc content (ppm) | 0.87 (0.38)[b] | 2.01 (1.23)[b] | 0.90 (0.68)[b] | 5.24 (3.02)[b] | 2.18 (1.86)[b] | 9.64 (5.26)[a] | <**0.001** |

Variables are listed in alphabetical order, values are means and standard deviations, and N is the total number of technical replicates within each treatment category. P-values are derived from linear mixed effects models conducted on each response variable, which were approximated using Type II Wald chisquare tests as implemented in the lme4 and car packages [28]. When omnibus tests were significant, pairwise comparisons were estimated using Tukey contrasts in the multcomp package [29].

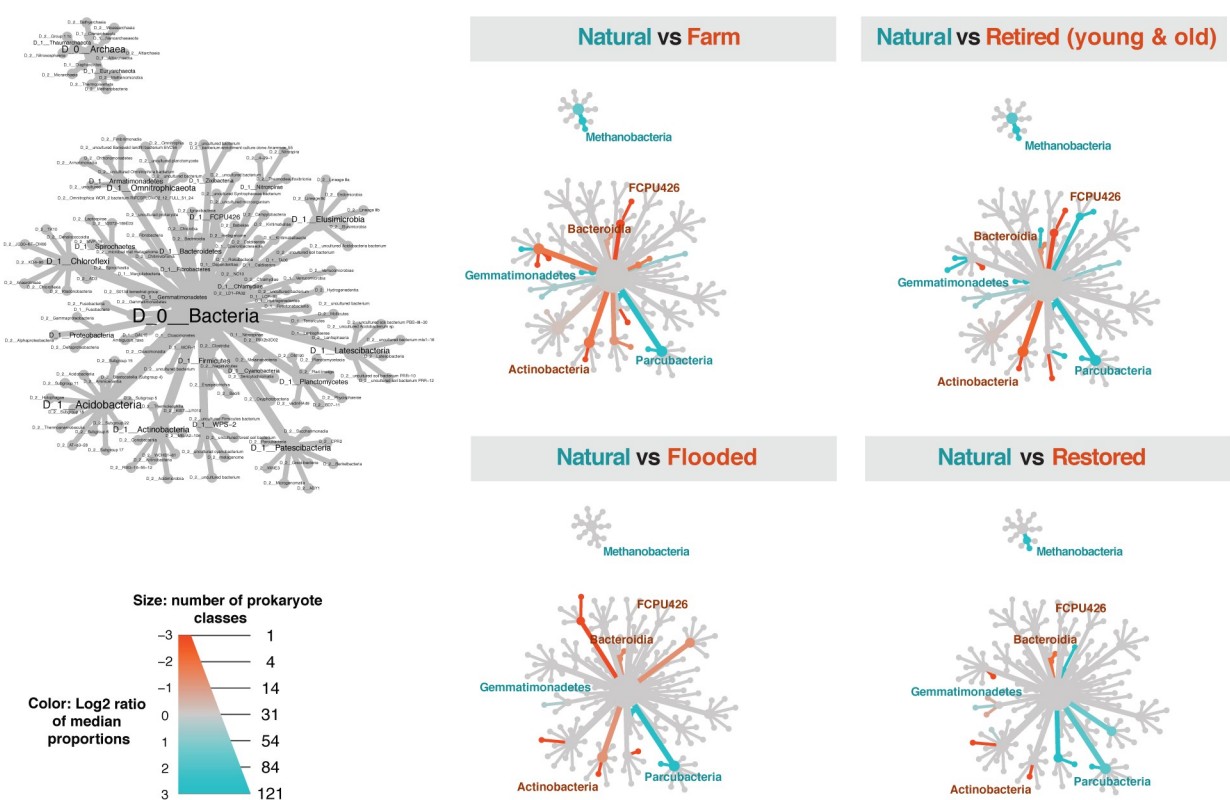

**Fig 4. Differences in prokaryotic taxa between treatments and natural wetlands.** Heat trees were constructed using the metacoder package [33], at the class taxonomic level. The size of the node in each cladogram is proportional to the number of unique classes within each phylum. Color intensity is proportional to the difference in abundance between natural sites and the other treatments, as calculated from the log2 ratio of median abundance. Young retired and old retired samples were pooled together for simplicity, since they had similar communities. Taxa that have a higher within-sample dominance in natural sites are shown in teal whereas taxa that have a higher within-sample dominance in farm, retired, flooded, and restored wetlands are shown in orange. Nonsignificant comparisons are shown in grey. Colored labels showcase example taxa that were consistently more (teal) or less (orange) abundant in natural sites or displayed variation across the four heat trees.

characteristics, even exceeding restoration in terms of soil organic matter accumulation and cation exchange capacity, after only four years. In contrast, retired sites failed to achieve natural wetland conditions, even after several decades following retirement. These results suggest that farms that are slated for retirement should be left in the flooded condition rather than the drained condition.

Spatial autocorrelation is often in inherent in observational studies, and should be considered in the interpretation of results. The Tidmarsh East (restored sites) were close in proximity to the natural sites so there is some uncertainty as to whether the patterns observed here are geographical artifacts or representative of restoration activities as a whole. Yet, while we found spatial autocorrelation (for the microeukaryote dataset only), it was unlikely to have affected our major conclusions: there was no relationship between geography and soil variables, but treatments clearly had a consistent effect on soil variables, suggesting that treatment effects overwhelmed intrinsic environmental gradients (**Fig 1**, **Table 1**). Likewise, the flooded sites were the furthest geographically from the natural sites, yet they often had the highest similarity to the natural sites, also indicating that the effect of treatments were much stronger than microbial dispersal limitation.

While some soil-based functions, such as respiration and soil organic matter formation are widespread among microbial taxa, other functions such methanogenesis and denitrification

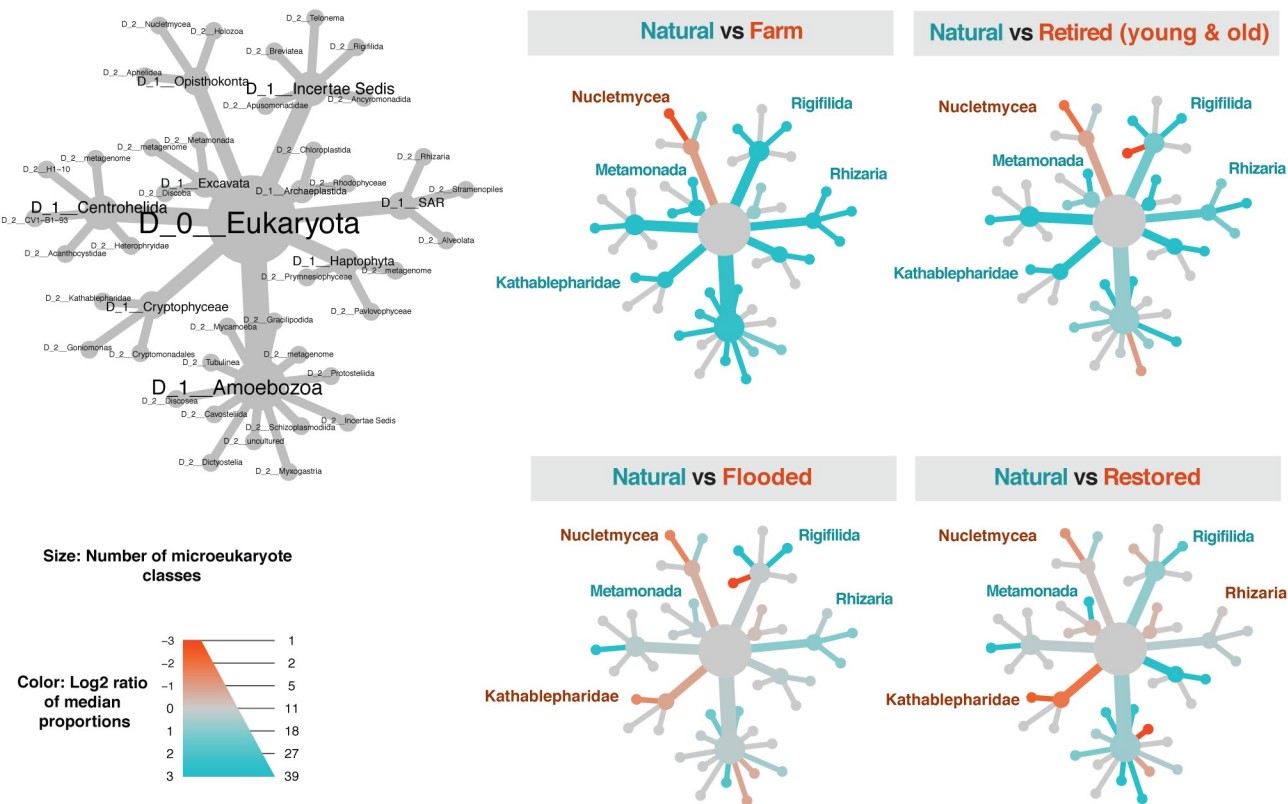

**Fig 5. Differences in eukaryotic taxa between management histories and natural wetlands.** Heat trees were constructed using the metacoder package, at the class level. The size of the nodes in each cladogram is proportional to the number of unique classes within each phylum, and color intensity is proportionate to the difference in abundance between natural sites and each of the other management categories, calculated from the log2 ratio of median abundance. Young retired and old retired samples were pooled together for simplicity, since they had similar communities. Taxa that have a higher within-sample dominance in natural sites are shown in teal whereas taxa that have a higher within-sample dominance in farm, retired, flooded, and restored wetlands are shown in orange. Nonsignificant comparisons are shown in grey. Colored labels showcase example taxa that were consistently more (teal) or less (orange) abundant in natural sites or displayed variation across the four heat trees.

may be narrowly distributed. In our study we found a strong correlation between microbial communities and soil functions; half of the variation in prokaryotes and microeukaryote communities could be explained by soil variables, including cation exchange capacity and denitrification. It has been previously found that soils with high denitrification potential also have a discrete microbial community [11, 34, 35]. In our study, denitrification was among the top six predictors for both prokaryotes and microeukaryotes, suggesting that the microeukaryotes surveyed here are contributing to denitrification themselves [36], or that microeukaryote communities sort with specific denitrifying bacterial communities. Furthermore, high methane fluxes in natural sites were associated with the dominance of *Methanobacteria*, a class of archaea containing several methanogens. These results lend further support to the idea that microbial taxa (or taxonomic groups) play deterministic roles in soil functions that are important to restoration practitioners, rather than being redundant to one another.

Microbial communities sorted with denitrification and cation exchange capacity, variables that restoration seeks to maximize, as well as underlying chemical gradients; prokaryote communities aligned with potassium base saturation, sodium, zinc and iron, and microeukaryote communities aligned with potassium base saturation, sodium, pH and copper. The high potassium base saturation in farm sites and retired sites may reflect a legacy of fertilizer application,

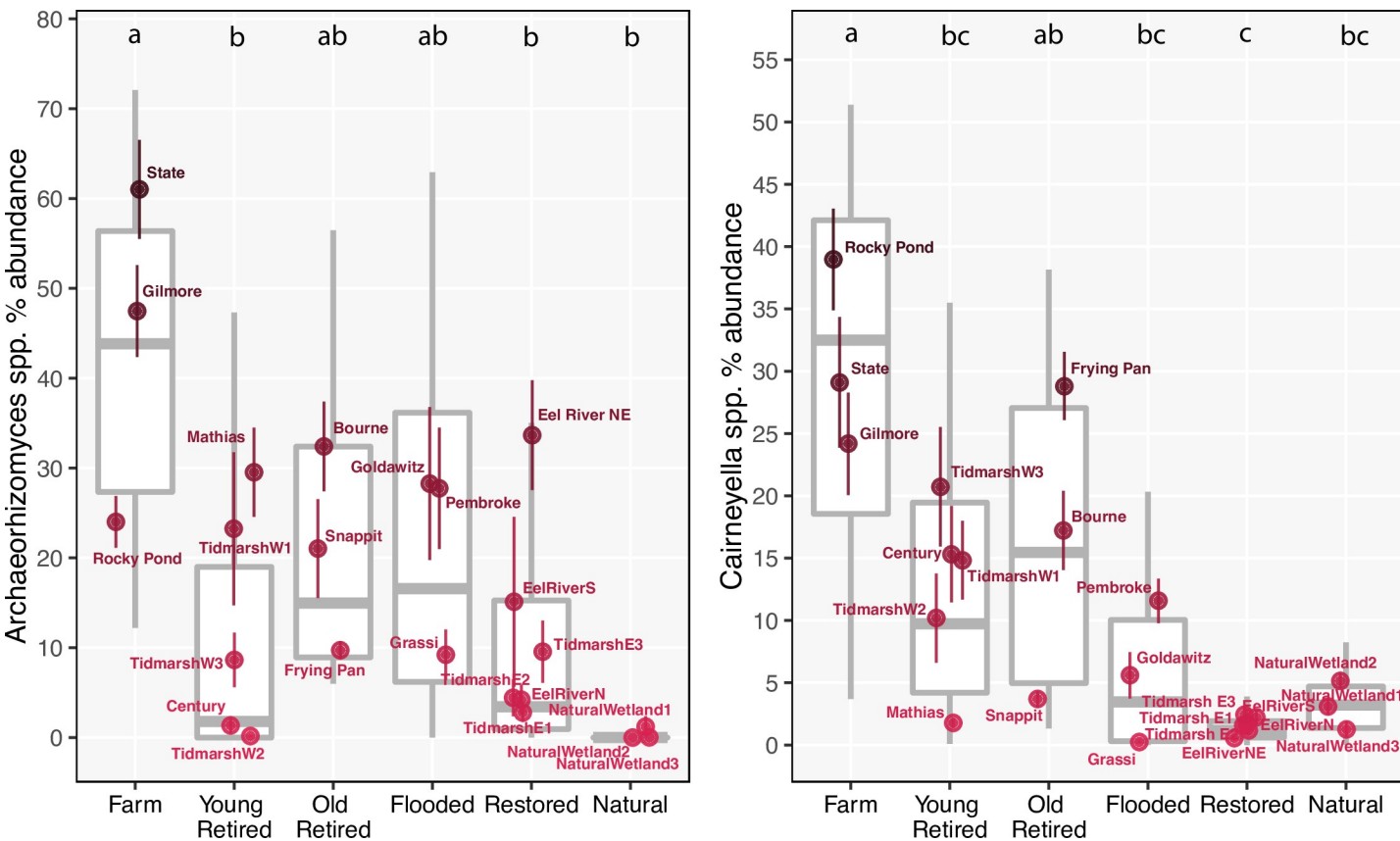

**Fig 6. Relative dominance of two most common eukaryotic genera.** The two most common eukaryotic genera were *Archaeorhizomyces* and *Cairneyella*, suspected mycorrhizal symbionts of the cranberry crop. Both genera had lower median within-sample dominance in young retired farms, old retired farms, flooded, restored, and natural sites than in cranberry farms. Points indicate mean abundance for each site and vertical lines are standard errors calculated from eight technical replicates. Grey box plots indicate the median and interquartile range of all soil samples within each treatment category.

whereas higher amounts of iron and zinc in flooded, restored and natural sites may be a direct result of the greater amount of soil organic matter found in these soils [37]. Natural sites were also saltier and more acidic than the other treatments, both of which are known environmental filters of microbial communities [38, 39]. Taken together, these results suggest that the effects of restoration treatments on microbial communities can be mediated by indirect effects on soil chemistry.

Differences in taxa dominance across treatments provided context for interpreting differences in biogeochemical variables. For instance, the high within-sample dominance of the methanogen *Methanobacteria* in natural and flooded sites makes sense, given that these sites both had saturated soils and some standing water. It has been previously noted that *Euryarchaeota*, a phylum containing methanogens, dominates wet ecosystems whereas *Thaumarchaeota*, a phylum containing ammonia oxidizers, dominates in drier systems [40]. Our finding that *Gemmatimonadetes* is a possible indicator of wetland development was supported by [41], which found *Gemmatimonadetes* in vegetated, but not bare wetlands. *Actinobacteria* and *Bacteroidia*, which were consistently less dominant in natural sites compared with the other treatments, may be indicators of disturbance or early succession, as noted by [42]. There are fewer ecological data sets on wetland microeukaryotes to compare our results to, but we note that *Rhizaria*, a large group that includes predators of both bacteria and autotrophic protists [43], was also identified as possible indicator of wetland development in our study. Longitudinal

studies over time could reveal early, mid, and late successional indicator communities, and complementary isotope labeling studies could provide insights into trophic interactions [44].

Cranberries are a unique agricultural crop because they are a native wetland perennial, grown in monocultures. We detected a belowground legacy of cranberry cultivation on the soil microbiome, findings which are echoed in investigations of *Vaccinium* patches in natural bogs [45, 46]. The two most abundant genera across the microeukaryote dataset, *Archaeorhizomyces spp.* and *Cairneyella spp.*, are suspected ectomycorrhizal and ericoid mycorrhizal symbionts of the cranberry crop [47, 48]. The decline of these genera in restored sites in particular was likely due to soil disturbance, uprooting of cranberry vines, and loss of host plants. Previous vegetation surveys in ecologically restored, former farms indicate that native vegetation establishes quickly following restoration, replacing most cranberry plants [10, 12, 49]. While it appears that these fungal symbionts track with aboveground cranberry abundance, future work evaluating rhizosphere community assembly around controlled plantings of cranberry and native wetland plants could reveal whether soil microbes control aboveground plant succession, or vice versa.

In previous studies, we have described the desirable impacts that active ecological restoration of retired cranberry farms has on both biogeochemical characteristics [50, 51] and microbial community structure [11]. Here, we found that flooding also promotes beneficial wetland functions, even exceeding wetland restoration for soil organic matter formation and cation exchange capacity. While flooding clearly promotes wetland biogeochemistry, flooding does not meet other restoration goals, such as repairing stream connectivity for fish and other aquatic life. Flooding is achieved by leaving artificial dams and water control structures in place, which necessarily impedes the flow of any streams or rivers on site. Many coastal cranberry farms are located on historically important migration and spawning grounds for anadromous fish such as alewife, blueback herring, rainbow smelt, American shad, and white perch. Restoration of these migratory habitats is one of the explicit primary goals of ecological restoration, and these goals are definitely not met when impoundments are left in place.

In conclusion, flooded and restored sites approached soil attributes of natural wetland sites, including prokaryote communities, microeukaryote communities, and 35 key soil variables. Effects were surprisingly strong for flooded sites, likely due to wetter conditions that support soil organic matter accumulation. While these results underscore the foundational role that water retention plays on soil development, we do not suggest that flooding should be a substitute for active ecological restoration. Rather, retired farms should be flooded during the restoration waiting period, or when ecological restoration is not an option.

## Supporting information

**S1 Data. This file is an excel spreadsheet containing site metadata, the 35 soil variables, and sequence counts for each sample.**
(XLSX)

**S1 File. This file contains S1 and S2 Figs as well as detailed methods for the soil variable analyses.**
(DOCX)

## Acknowledgments

We thank Glorianna Davenport, Evan Schulman, Alex Hackman, Helen Castles, Martha Sylvia, Mass Audubon, and several landowners for field access and support. The Cary Institute of Ecosystem Studies provided technical assistance for soil analyses. This work was conducted as

part of the Living Observatory and the Mount Holyoke Restoration Ecology Program, in collaboration with the MA DER Cranberry Bog Program. We thank the hosts of the 2020 QIIME2 Workshop in Bethesda, MD, and Rachael Lappan for the great 16S and 18S QIIME2 tutorial. Finally, we acknowledge the Wampanoag Nation, and their ancestral homeland on which this survey was conducted.

## Author Contributions

**Conceptualization:** Rachel L. Rubin, Kate A. Ballantine, Jason P. Andras.

**Data curation:** Rachel L. Rubin, Kate A. Ballantine, Arden Hegberg, Jason P. Andras.

**Formal analysis:** Rachel L. Rubin.

**Funding acquisition:** Kate A. Ballantine, Jason P. Andras.

**Investigation:** Rachel L. Rubin, Kate A. Ballantine, Arden Hegberg, Jason P. Andras.

**Methodology:** Rachel L. Rubin, Kate A. Ballantine, Jason P. Andras.

**Project administration:** Kate A. Ballantine, Jason P. Andras.

**Resources:** Rachel L. Rubin, Kate A. Ballantine, Jason P. Andras.

**Software:** Rachel L. Rubin.

**Supervision:** Rachel L. Rubin, Kate A. Ballantine, Jason P. Andras.

**Validation:** Rachel L. Rubin, Kate A. Ballantine, Jason P. Andras.

**Visualization:** Rachel L. Rubin.

**Writing – original draft:** Rachel L. Rubin, Kate A. Ballantine, Arden Hegberg, Jason P. Andras.

**Writing – review & editing:** Rachel L. Rubin, Kate A. Ballantine, Arden Hegberg, Jason P. Andras.

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
