## [Decision Letter · Decision Letter 0]

10 Dec 2020

PONE-D-20-33816

If you flood it, they will come: prokaryote and microeukaryote communities in passively restored wetlands approach reference conditions

PLOS ONE

Dear Dr. Rachel Rubin,

Thank you for submitting your manuscript to PLOS ONE. After careful consideration, we feel that it has merit but does not fully meet PLOS ONE’s publication criteria as it currently stands. Therefore, we invite you to submit a revised version of the manuscript that addresses the points raised during the review process.

We look forward to receiving your revised manuscript.

Kind regards,

Ashwani Kumar, Ph.D.

Academic Editor

PLOS ONE

Additional Editor Comments (if provided):

Dear Authors,

I have received the response from the reviewers and they recommended this manuscript for revision. Please revise in track change carefully with point to point answers to their queries. I would like to see the revise manuscript.

Thanks

Sincerely

ashwani

Journal Requirements:

2) Thank you for stating the following in the Acknowledgments Section of your manuscript:

[Funders include the Department of Energy Joint Genome Institute (Project 1188787) and the Mount Holyoke Restoration Ecology Program.]

 [Joint Genome Institute Project 1188787]

3) Thank you for stating the following in your Competing Interests section: 

[NO].

4) We note that Figure 1 in your submission contain [map/satellite] images which may be copyrighted. All PLOS content is published under the Creative Commons Attribution License (CC BY 4.0), which means that the manuscript, images, and Supporting Information files will be freely available online, and any third party is permitted to access, download, copy, distribute, and use these materials in any way, even commercially, with proper attribution. For these reasons, we cannot publish previously copyrighted maps or satellite images created using proprietary data, such as Google software (Google Maps, Street View, and Earth). For more information, see our copyright guidelines: http://journals.plos.org/plosone/s/licenses-and-copyright.

i.    You may seek permission from the original copyright holder of Figure(s) [#] to publish the content specifically under the CC BY 4.0 license.

ii.    If you are unable to obtain permission from the original copyright holder to publish these figures under the CC BY 4.0 license or if the copyright holder’s requirements are incompatible with the CC BY 4.0 license, please either i) remove the figure or ii) supply a replacement figure that complies with the CC BY 4.0 license. Please check copyright information on all replacement figures and update the figure caption with source information. If applicable, please specify in the figure caption text when a figure is similar but not identical to the original image and is therefore for illustrative purposes only.

Reviewers' comments:

Reviewer's Responses to Questions

**Comments to the Author**

1. Is the manuscript technically sound, and do the data support the conclusions?

Reviewer #1: No

Reviewer #2: Yes

2. Has the statistical analysis been performed appropriately and rigorously? 

Reviewer #1: No

Reviewer #2: Yes

3. Have the authors made all data underlying the findings in their manuscript fully available?

Reviewer #1: Yes

Reviewer #2: No

4. Is the manuscript presented in an intelligible fashion and written in standard English?

Reviewer #1: Yes

Reviewer #2: Yes

5. Review Comments to the Author

Reviewer #1: PONE-D-2033816

The manuscript is generally written very clearly and concisely. I am very grateful to the authors for that! The subject matter is also very interesting. Unfortunately, as the manuscript currently reads, I cannot recommend publication because 1) the analyses reflect pseudoreplication, 2) the soil sampling may have been flawed, and 3) age differences among sites of different treatments may prohibit comparisons among treatments.

Pseudoreplication

Lines 118-123. As far as I can tell, there is a single reference wetland, 2 actively restored sites, 3 passively restored sites, an unknown number of retired sites and an unknown number of active farms. The authors need to be clearer about replication for each treatment. This information and site descriptions should probably be made available in a table. Despite the limited replication at the site level, each individual soil sample was apparently analyzed as if it were independent of the other samples. I do not see how the authors can justify this pseudoreplication. The unit of replication is the SITE, not the SOIL SAMPLE because the treatments were applied at the level of the site, not the level of the soil sample. Therefore, these analyses are incorrect.

Soil sampling

Line 140. If the ratio of muck to sand varies from site to site, then one will probably see corresponding variation in microbial community structure, SOM concentration, CEC, etc. that has more to do with the relative amounts of muck and sand rather than anything to do with restoration effort. In other words, the properties of the muck may be highly similar in ALL sites, and the differences and similarities among sites may be explained simply by the amount of sand vs. muck in the soil samples, not necessarily by the level of restoration.

Age effects

Lines 118 – 123. Obviously, it is difficult to find retired farms, passively restored sites, and actively restored sites that are similar to each other except in the way they have been managed following farm retirement. But clearly time may affect many of the variables that were assessed. There are 3 replicate passively restored sites of the same age (4 y), but the actively restored sites were 1 and 6 years old, and the retired farms were newly retired (1-4 y) or “old retired”: (17-20 y). Which of the active farms, which of the actively restored sites, which of the retired farms, and which of the passively restored sites can logically be compared to the reference wetland? The authors will have to take age into consideration in drawing conclusions. On line 244, the authors remove site age as a “nuisance variable”. I do not understand the justification for doing so. I do not see how you can remove the effect of age from among, for example, the actively restored sites, of which there are only two. Presumably every variable is modeled as a linear function of age, but how can that be justified if there are only two replicates of given treatment? Moreover, there is but a single reference wetland. Such a justification is possible only if each soil sample is considered an independent variable, and I have already indicated above why this is not so.

Smaller editorial comments:

Line 74. The word “ditches” is used twice.

Line 102. It is not immediately clear what the difference is between retired farms and flooded sites because flooded sites are presumably retired farms.

Lines 101 – 103. Are these really the only questions? Certainly, these are important but, judging from the rest of the introduction, an important underlying management question concerns the effectiveness of passive flooding as a restoration strategy. Shouldn’t that be included?

Line 106. Yes, but what basis is there for thinking that simply because “extensive action” was taken, active restoration would be the best strategy? Is there any evidence for this expectation? One could possibly propose another even more logical, opposing hypothesis, which is that active restoration would result in less SOM accumulation because of the greater level of disturbance.

Line 115. Describing the length of the perimeter is not helpful at all unless we know the shape.

Lines 128-129. Previously the authors used the term “flashboard”, but here they use the term “wooden board”. Keep it simple and use just one term.

Line 141. “…at a depth of 10 cm” is not clear. Do the authors mean “…to a depth of 10 cm”?

Line 141. What does “Soil samples (50 cm3)” mean? Does this mean that 50 cm3 samples were taken at 10 cm depth?

Line 142. Were all 8 samples homogenized together, or do you mean each of the 8, 50 cm3 samples was homogenized?

Reviewer #2: The manuscript presented by Rubin et al is an interesting example of how restoration processes have effects on microbial soil communities. The paper is well written, and I think it is easy to understand. I agree with the conclusions obtained here; however, I have some concerns about the data showed here and the data that the authors did not show.

Please, show at least in supplementary data, a table containing number of raw reads, reads per sample after filtering, taxonomy… those data showed in most of papers, that I believe is mandatory.

Authors refer ASVs correspond to unique taxonomic assignments to genus and species level for 16S (lines 173-175). Which genera within the identified Classes are the most abundant? Same question remains unanswered for microeukaryotic populations.

How the authors removed chimeras and sequences from plants/algae in the 16S analyses?

Some minor comments:

Line 74, remove one ditches

Line 149-154, please identify the regions for 16S and 18S (V3-V5, V4…)

The elimination of the R2 raw reads represents here an issue. Please, explain pros and cons of using only R1 or give more reasons/discussion about how this affects the data quality. This is a limitation of the study.

Data regarding to sequencing must be presented at least as supplementary data.

Line 181, Charophytes refers only to green algae, why the authors remove this taxon if they form part of the microbial communities as a whole?

Please, upload raw reads to SRA NCBI for better data accessibility. MG-RAST showed no results for the number provided.

6. PLOS authors have the option to publish the peer review history of their article (what does this mean?). If published, this will include your full peer review and any attached files.

Reviewer #1: No

Reviewer #2: No

---

## [Author Response · Author response to Decision Letter 0]

23 Jan 2021

Comments are provided in the packet, together with the cover letter.

---

## [Decision Letter · Decision Letter 1]

7 Apr 2021

PONE-D-20-33816R1

Flooding and ecological restoration promote wetland microbial communities and soil functions on former cranberry farmland

PLOS ONE

Dear Dr. Rachel Rubin,

Thank you for submitting your manuscript to PLOS ONE. After careful consideration, we feel that it has merit but does not fully meet PLOS ONE’s publication criteria as it currently stands. Therefore, we invite you to submit a revised version of the manuscript that addresses the points raised during the review process.

We look forward to receiving your revised manuscript.

Kind regards,

Luigimaria Borruso

Academic Editor

PLOS ONE

Journal Requirements:

Reviewers' comments:

Reviewer's Responses to Questions

**Comments to the Author**

1. If the authors have adequately addressed your comments raised in a previous round of review and you feel that this manuscript is now acceptable for publication, you may indicate that here to bypass the “Comments to the Author” section, enter your conflict of interest statement in the “Confidential to Editor” section, and submit your "Accept" recommendation.

Reviewer #1: (No Response)

Reviewer #3: (No Response)

2. Is the manuscript technically sound, and do the data support the conclusions?

Reviewer #1: Partly

Reviewer #3: Yes

3. Has the statistical analysis been performed appropriately and rigorously? 

Reviewer #1: I Don't Know

Reviewer #3: Yes

4. Have the authors made all data underlying the findings in their manuscript fully available?

Reviewer #1: Yes

Reviewer #3: Yes

5. Is the manuscript presented in an intelligible fashion and written in standard English?

Reviewer #1: Yes

Reviewer #3: Yes

6. Review Comments to the Author

Reviewer #1: PONE-D-20-33816R1

In general, this manuscript is very much improved over the first iteration. My main concern is Table 1, which appears to list the wrong values for n. I hope it is just an oversight and that the authors did not really perform incorrect analyses. In addition, the following are some other things to consider.

Line 68. “Data” is a plural word. Thus, “data that are…” is correct.

Lines 99-101. I do not understand the reasoning here. If flooding produces comparable results to restoration, why use flooding merely as an intermediate step, or as an alternative only when restoration is NOT an option? If it is comparable, why not use it ALL the time, particularly if it is easier, faster and cheaper than restoration?

Line 120. I do not seem to have access to the S1 file. I looked for this file because I do not see a good description of the natural sites. In fact, they are not described as wetlands in the text, only that “natural reference sites have no history of cranberry farming (Fig 1).”

Line 144. What does this mean: “For the MAJORITY of cases”? It might be better to state how many cases out of how many cases so the reader can make the judgement as to whether the procedure justifies what was done.

Lines 144-146. This sentence seems to be missing something.

Line 154. “spaced 30 m apart” is not clear. In a straight line? Often non-random sampling schemes like this are not easily justified. The point is for the sampling to best represent the kind of real variability likely to exist at the site.

Line 155. What kind of sampling device was used? It seems like a cylindrical soil probe was used, in which case the width should be given as the diameter.

Lines 233-234. How did you control for spatial autocorrelation?

Lines 234-235. It is probably more accurate to write “spatial correlation is unavoidable in many ecological studies”. But this still does not eliminate the potential problems interpreting this dataset. The interpretations have to take this potential confounding into consideration.

Line 242. Explain ASV.

Table 1. I do not understand the values for n. The unit of replication is the site, not the soil sample. Therefore, I am not confident that the statistical analyses were done properly.

Reviewer #3: The present article investigates the effects of retiring cranberry farms, and compares how retiring alone, flooding and active restoration are affecting soil microbial communities.

Although numbers of real replicates are low, appropriate efforts such as statistical power tests have been applied to compensate for this drawback. Moreover, individual soils have been sampled intensively to account for within-soil heterogeneity. Certainly, it is not easy to find and sample suitable sites for these five(!) treatments. The ecological conclusions are important for restoration politics and can contribute to a site melioration during the time leading up to restoration.

I do have some minor points that should be addressed before publication:

INTRODUCTION

Do soil variables explain microbial communities or vice versa? You mentioned this point that is especially true for denitrification in the discussion (l445) but I suggest to pronounce it more in the introduction.

l58: reference is missing

METHODS

l144: Unfortunately, FigS2 doesn’t show any data, at least in my file.

l151: please include the projection type that you used in this map.

l191: At this point, the reader is not familiar with the identifier M3. I suggest either to reference the position where these identifiers are explained, or to shortly state what kind of site it was. e.g. restored site M3

I 201: Being new to this method, I would appreciate some more information on how this extraction works in the main text.

l205: Please include a reference for your spectrometry.

l241: Paradis and Schleip 2018 is not included in the reference list

l263-264: this statement is not easy to understand. Above you stated that your environmental data didn’t show a relationship between geography and soil variables but here you mention short environmental gradients. Please explain more carefully.

RESULTS

l316 + l 337(+figure caption of fig 3): Adding up both percentages in fig. 3A/B I am not getting the 45% /44% mentioned in the text (36.5/28%). Whether this is due to the forward selection of paricular variables or due to other reasons, please explain more carefully.

Table S1 is missing many values for sample ER.2.5: It is not clear, however, if this is a real artefact, or if it is only missing by mistake. If values were not obtained for this sample, plese state something like n.d. in the excel file.

l350-352: In order to complete this information, I suggest to include two columns indicating the importance of each variable for microeucariotes/procaryotes, and to shorten the table caption accordingly.

Fig. 4: I really like those graphs, they are very informative! However, you should stay consistent in what you are labelling and what you don’t. Please include Methanobacteria labeling that is missing in the graph depicting differences between natural and restored areas. Other labels of significant differences are also missing (same in fig. 5) and I am wondering why. Were they not as important or did you choose to label only the once with a certain log2 ratio of median proportions?

l379: …whereas taxa that are more abundant in retired, flooded, and restored… here, farm sites are missing from the description.

DISCUSSION

l391-392: Wasn’t this class (kathablepharidae) MORE abundant in flooded and restored sites (orange) as compared to natural sites? If this is the case, they seem to represent a species that gains importance in succession stages such as flooded and restored sites, but is not as important in the natural reference sites. Please correct also in the discussion.

Fig. 5: Nucletmycea are labelled as Nucleomycea in some graphs, please correct.

l408: This procedere has not been described in the methods. How did you define “standouts”?

7. PLOS authors have the option to publish the peer review history of their article (what does this mean?). If published, this will include your full peer review and any attached files.

Reviewer #1: No

Reviewer #3: **Yes: **Magdalena Nagler

---

## [Decision Letter · Decision Letter 2]

30 Jun 2021

PONE-D-20-33816R2

Flooding and ecological restoration promote wetland microbial communities and soil functions on former cranberry farmland

PLOS ONE

Dear Dr. Rachel Rubin,

Thank you for submitting your manuscript to PLOS ONE. After careful consideration, we feel that it has merit but does not fully meet PLOS ONE’s publication criteria as it currently stands. Therefore, we invite you to submit a revised version of the manuscript that addresses the points raised during the review process.

We look forward to receiving your revised manuscript.

Kind regards,

Luigimaria Borruso

Academic Editor

PLOS ONE

Journal Requirements:

Reviewers' comments:

Reviewer's Responses to Questions

**Comments to the Author**

1. If the authors have adequately addressed your comments raised in a previous round of review and you feel that this manuscript is now acceptable for publication, you may indicate that here to bypass the “Comments to the Author” section, enter your conflict of interest statement in the “Confidential to Editor” section, and submit your "Accept" recommendation.

Reviewer #1: (No Response)

Reviewer #3: All comments have been addressed

2. Is the manuscript technically sound, and do the data support the conclusions?

Reviewer #1: No

Reviewer #3: Yes

3. Has the statistical analysis been performed appropriately and rigorously? 

Reviewer #1: No

Reviewer #3: Yes

4. Have the authors made all data underlying the findings in their manuscript fully available?

Reviewer #1: Yes

Reviewer #3: Yes

5. Is the manuscript presented in an intelligible fashion and written in standard English?

Reviewer #1: Yes

Reviewer #3: Yes

6. Review Comments to the Author

Reviewer #1: PONE-D-20-33816R2

I believe we have a fundamental disagreement about the unit of replication in this study. The authors believe that the soil sample is the unit of replication. Thus, in Table 1, n = 24 or n = 48. On the other hand, I believe the unit of replication is the farm or the flooded site or the restored site, etc. The reason for this is that the unit of replication is the unit to which the treatment is applied. Individual soil samples from a single farm do not represent units of replication simply because the treatment was not individually applied to each soil sample. Therefore, I must insist that the authors perform their statistical analyses correctly.I mentioned this in my last review.

Reviewer #3: The authors addressed most of my comments. The only critcial point, which may be addressed, however, by the publishing team is that FIG S2 in file S2 still doesn’t display any data.

L62: I have the feeling that “are” needs to be removed

L449: The effect of treatments was stronger,

L452: isn’t it the S2 File?

7. PLOS authors have the option to publish the peer review history of their article (what does this mean?). If published, this will include your full peer review and any attached files.

Reviewer #1: No

Reviewer #3: **Yes: **Magdalena Nagler, PhD

---

## [Decision Letter · Decision Letter 3]

18 Oct 2021

PONE-D-20-33816R3Flooding and ecological restoration promote wetland microbial communities and soil functions on former cranberry farmlandPLOS ONE

Dear Dr. Rachel Rubin,

Thank you for submitting your manuscript to PLOS ONE. After careful consideration, we feel that it has merit but does not fully meet PLOS ONE’s publication criteria as it currently stands. Therefore, we invite you to submit a revised version of the manuscript that addresses the points raised during the review process. I gently ask you to consider the comments regarding the statistics approach the review#I arose since the objection from her/his original review still stands (see below).

We look forward to receiving your revised manuscript.

Kind regards,

Luigimaria Borruso

Academic Editor

PLOS ONE

Reviewers' comments:

Reviewer's Responses to Questions

**Comments to the Author**

1. If the authors have adequately addressed your comments raised in a previous round of review and you feel that this manuscript is now acceptable for publication, you may indicate that here to bypass the “Comments to the Author” section, enter your conflict of interest statement in the “Confidential to Editor” section, and submit your "Accept" recommendation.

Reviewer #1: (No Response)

Reviewer #3: All comments have been addressed

2. Is the manuscript technically sound, and do the data support the conclusions?

Reviewer #1: No

Reviewer #3: Yes

3. Has the statistical analysis been performed appropriately and rigorously? 

Reviewer #1: No

Reviewer #3: Yes

4. Have the authors made all data underlying the findings in their manuscript fully available?

Reviewer #1: Yes

Reviewer #3: Yes

5. Is the manuscript presented in an intelligible fashion and written in standard English?

Reviewer #1: Yes

Reviewer #3: Yes

6. Review Comments to the Author

Reviewer #1: I am not satisfied that the authors have handled the statistical analyses correctly. The objection from my original review two iterations ago still stands.

Reviewer #3: The authors addressed all issues raised in the former revision and I am happy with how the manuscript improved. Before publication, fig. S2 needs to be re-formatted as in my version I can see the graphs with x- and y-axes, but those graphs are not displaying any data.

7. PLOS authors have the option to publish the peer review history of their article (what does this mean?). If published, this will include your full peer review and any attached files.

Reviewer #1: No

Reviewer #3: **Yes: **Magdalena Nagler

---

## [Author Response · Author response to Decision Letter 3]

15 Nov 2021

Thank you for your prompt reply regarding our manuscript, PONE-D-20-33816R3, "Flooding and ecological restoration promote wetland microbial communities and soil functions on former cranberry farmland". We are hoping you can provide some additional guidance in your role as editor to help us move our manuscript forward to publication. 

As you have noted, the sole remaining point of contention regarding our manuscript is the objection of Reviewer 1 to aspects of our statistical analysis. We have now made substantive changes in multiple rounds of revision that directly address Reviewer 1's concerns, and we have provided detailed commentary detailing how our revisions resolve those concerns. Specifically, Reviewer 1 asserted that our analysis suffered from pseudoreplication, since it combined samples across sites into treatment groups. Though we were not in complete agreement with this assessment, we acquiesced and in our most recent revision ran a new mixed model that incorporates "site" as a random effect in the statistical model. Furthermore, we also re-graphed Fig 2 and Fig 6 to display mean values for each site, with technical replicates shown as error bars. These revisions directly and definitively address Reviewer 1's concern. Moreover, beyond our own assertion of the validity of our analysis, we now also have the clear approval of Reviewer #3 who states that, "The authors addressed all issues raised in the former revision and I am happy with how the manuscript improved." In contrast, Reviewer 1 provides only a brief and vague objection, with no commentary at all regarding our prior revisions or our explanation of those revisions and no specific request for changes or corrections. Such a review is not actionable and therefore of no practical use. 

We respectfully note that, in the course of peer review, it may sometimes be the case that opinions will differ, and not every Reviewer will be completely satisfied. In such cases, it seems reasonable to call upon the discretion of the Editor to make a judgement or a specific suggestion for resolution, and we expect that you will do this now. Given our substantive revisions, our detailed explication of those revisions, and the specific and definitive endorsement of the other independent Reviewer, can you please either accept our manuscript or tell us what specific change(s) you would like to see to render it suitable for publication? 

Signed, Jason Andras (corresponding author)

---

## [Editor Report · Decision Letter 4]

22 Nov 2021

Flooding and ecological restoration promote wetland microbial communities and soil functions on former cranberry farmland

PONE-D-20-33816R4

Dear Jason,

We’re pleased to inform you that your manuscript has been judged scientifically suitable for publication and will be formally accepted for publication once it meets all outstanding technical requirements.

Kind regards,

Luigimaria Borruso

Academic Editor

PLOS ONE
---

## [Editor Report · Acceptance letter]

9 Dec 2021

PONE-D-20-33816R4 

Flooding and ecological restoration promote wetland microbial communities and soil functions on former cranberry farmland 

Dear Dr. Rubin:

I'm pleased to inform you that your manuscript has been deemed suitable for publication in PLOS ONE. Congratulations! Your manuscript is now with our production department. 

Kind regards, 

on behalf of

Dr. Luigimaria Borruso 

Academic Editor

PLOS ONE